# Clinical Management of Diabetes Mellitus in the Era of COVID-19: Practical Issues, Peculiarities and Concerns

**DOI:** 10.3390/jcm9072288

**Published:** 2020-07-18

**Authors:** Chrysi Koliaki, Anastasios Tentolouris, Ioanna Eleftheriadou, Andreas Melidonis, George Dimitriadis, Nikolaos Tentolouris

**Affiliations:** 1First Department of Propaedeutic Internal Medicine and Diabetes Center, Medical School, National Kapodistrian University of Athens, Laiko General Hospital, 11527 Athens, Greece; ckoliaki@yahoo.com (C.K.); antentol@med.uoa.gr (A.T.); joeleftheriadou@yahoo.com (I.E.); 2Hellenic Diabetes Association, 11528 Athens, Greece; melidonisa@yahoo.com (A.M.); gdimitr@med.uoa.gr (G.D.); 3Cardiometabolic Department, Metropolitan Hospital, 18547 Neo Faliro, Greece; 4Second Department of Internal Medicine and Research Institute, Medical School, National and Kapodistrian University of Athens, Attikon University General Hospital, 12462 Chaidari, Greece

**Keywords:** coronavirus disease 2019 (COVID-19), severe acute respiratory distress syndrome coronavirus 2 (SARS-CoV2), diabetes mellitus, glycemic control, antidiabetic drugs, severity, mortality

## Abstract

The management of patients with diabetes mellitus (DM) in the era of the COVID-19 pandemic can be challenging. Even if they are not infected, they are at risk of dysregulated glycemic control due to the restrictive measures which compromise and disrupt healthcare delivery. In the case of infection, people with DM have an increased risk of developing severe complications. The major principles of optimal care for mild outpatient cases include a patient-tailored therapeutic approach, regular glucose monitoring and adherence to medical recommendations regarding lifestyle measures and drug treatment. For critically ill hospitalized patients, tight monitoring of glucose, fluids, electrolytes, pH and blood ketones is of paramount importance to optimize outcomes. All patients with DM do not have an equally increased risk for severity and mortality due to COVID-19. Certain clinical and biological characteristics determine high-risk phenotypes within the DM population and such prognostic markers need to be characterized in future studies. Further research is needed to examine which subgroups of DM patients are expected to benefit the most from specific antiviral, immunomodulatory and other treatment strategies in the context of patient-tailored precision medicine, which emerges as an urgent priority in the era of COVID-19.

## 1. Introduction

Since January 2020, the entire world has been facing an unprecedented healthcare crisis, namely the rapidly evolving outbreak of coronavirus disease 2019 (COVID-19) caused by the severe acute respiratory distress syndrome coronavirus 2 (SARS-CoV2). COVID-19 was declared a global pandemic by the World Health Organization (WHO) on 11th March 2020 [1], and has already threatened and challenged societies and healthcare systems all around the world. At the time of writing, more than 9 million people in more than 200 countries and territories have been infected and more than 450,000 deaths have been ascribed to COVID-19 [2]. The United States of America (USA) and certain European countries have experienced a dramatically high disease burden [2]. In Greece, the number of confirmed infected cases and deaths has remained relatively low, possibly due to the early implementation of strict infection control measures, such as generalized lockdown and school closure.

COVID-19 is a highly transmissible infection which leads to a broad spectrum of clinical manifestations, ranging from a mild flu-like illness to severe bilateral pneumonia, acute respiratory distress syndrome (ARDS), septic shock and multi-organ failure [3]. The COVID-19 pandemic affects predominantly socioeconomically deprived populations. Beyond age- and sex-related differences, there are significant racial and ethnic disparities in COVID-19 health outcomes, with non-Hispanic Afro-American black minorities being disproportionately affected in terms of COVID-19 incidence and mortality [4,5,6]. In multivariable analyses, black race and socioeconomic deprivation have been independently associated with an increased likelihood of hospital admission due to COVID-19 [7], but not independently associated with increased in-hospital mortality [7,8]. The accuracy of currently available molecular diagnostic tests for COVID-19 is not optimal [9]. Reverse transcriptase polymerase chain reaction tests (rt-PCR) are highly specific (specificity around 95%) but their sensitivity is only moderate and varies greatly depending on the timing and site of specimen collection, the quality of sampling, the stage of the disease and the degree of viral multiplication [10,11,12]. A systematic review of the accuracy of COVID-19 tests reported false-negative rates of between 2% and 29% (sensitivity equal to 71–98%) [13]. However, even with sensitivity values as high as 90%, the public health risk of false-negative results becomes substantial as the prevalence of COVID-19 rises [9]. With regard to treatment, patients with respiratory failure due to COVID-19 should be intubated and mechanically ventilated early in the disease course to slow disease progression and improve prognosis [14]. However, in the setting of overwhelmed intensive care units (ICUs) and a shortage of ventilators, alternative non-invasive ventilatory support options have been tested, including continuous positive airway pressure (CPAP) and high-flow nasal cannula (HFNC). These devices can temporarily alleviate respiratory failure, but they have been associated with an increased risk of nosocomial viral transmission to healthcare workers due to virus aerosolization [14]. WHO guidelines advocate the use of CPAP or HFNC to manage hypoxemic respiratory failure in patients with COVID-19, provided that medical staff wear appropriate personal protective equipment (PPE) [15].

Multiple datasets from China, Italy and the USA have consistently reported that the clinical course of COVID-19 is more severe in patients with advanced age (>70 years old) and pre-existing comorbidities, predominantly diabetes mellitus (DM), hypertension and cardiovascular disease [16,17,18]. The association between DM and COVID-19 is bidirectional. On the one hand, DM may increase the risk of contracting SARS-CoV2 and further complicate the clinical course of COVID-19, leading to increased severity and mortality [19,20]. On the other hand, SARS-CoV2 may directly attack pancreatic islets and either induce an acute insulin-dependent DM in previously non-diabetic subjects due to reduced insulin secretion or deteriorate glycemic control in subjects with pre-existing DM [21,22,23,24]. Upon SARS-CoV2 infection, patients with DM are highly likely to experience a significant deterioration of their glycemic control and ultimately need medical guidance to perform appropriate adjustments to their antidiabetic treatment to prevent uncontrolled hyperglycemia. Beyond the direct risks of infection, patients with DM, similar to patients with other chronic diseases, are further confronted with the considerable problem of suboptimal health service provision and the limited availability of medical resources related to the challenging circumstances of the pandemic. Under the pressure of COVID-19, healthcare systems are saturated and even overwhelmed, and the delivery of healthcare becomes interrupted and fragmented due to the high burden of infected cases [25]. Due to a potential lack of access to medications and supplies, patients with DM may experience difficulties in acquiring essential drugs or consumable devices, such as insulin pens, alcohol wipes and glucose test strips. Combined with suboptimal medical surveillance due to self-isolation and quarantine, this might lead to a metabolic decompensation and inadequate control of other DM-related comorbidities, such as hypertension and dyslipidemia, predisposing patients to adverse clinical outcomes [26].

The clinical management of patients with DM in the era of COVID-19 can be particularly challenging. The major general principles of optimal care for mild outpatient cases include a patient-tailored therapeutic approach, regular glucose monitoring and adherence to medical recommendations regarding lifestyle measures and drug treatment [27,28,29]. For critically ill hospitalized patients, especially in the setting of an ICU, tight glucose monitoring and the careful consideration of potential drug interactions are of paramount importance to optimize outcomes [29,30].

In the present review, we aimed to critically appraise the available literature regarding the association between DM and COVID-19, and mainly provide practical recommendations both for the prevention of COVID-19 in patients with DM, and for the optimal management of hyperglycemia in patients with DM and COVID-19, emphasizing the fundamental role of glucose monitoring and optimal glycemic control. We discuss the special precautions and considerations that need to be taken for patients with DM in the era of the COVID-19 pandemic, we outline special warnings and safety concerns for specific antidiabetic drug classes and we highlight knowledge gaps and unresolved issues which promptly warrant further investigation in order to advance our current understanding of the interaction between COVID-19 and DM and optimize outcomes for patients with DM.

## 2. Literature Search Strategy and Selection Criteria

The studies selected for this review were retrieved from the authors’ personal files or identified by a computer search program using the PubMed, Scopus and Web of Science electronic databases, searching for scientific literature published in English up to July 2020. Combinations of the following search terms were applied: “coronavirus”, “COVID-19”, “SARS-CoV2”, “diabetes mellitus”, “type 1 diabetes”, “type 2 diabetes”, “glycemic control”, “antidiabetic drugs” and “comorbidities”. Additional references were retrieved from reviewing the references cited in the original articles. The final reference list was generated on the basis of relevance to the topics covered in this publication, with the aim of covering multiple aspects of the association between DM and COVID-19, and providing practical recommendations for the prevention of COVID-19 in patients with DM, and for the optimal management of hyperglycemia in patients with DM and COVID-19.

## 3. The Association between DM and COVID-19

### 3.1. Causality

The relationship between COVID-19 and DM has been described as “the interaction of two pandemics”, since DM is the most common non-communicable chronic disease globally and, in parallel, is one of the major comorbidities in patients with COVID-19 [18,31]. Based on the current evidence, it remains controversial whether DM increases the susceptibility to SARS-CoV2 infection. Two meta-analyses of Chinese studies [18,32] and a preliminary report from the US Center for Disease Control and Prevention (CDC) [33] demonstrated that DM is a highly prevalent comorbidity in subjects with COVID-19 but has a prevalence comparable to that in the general population, suggesting that DM may not increase the risk of SARS-CoV2 infection. Studies in patients who were hospitalized [19] or died due to COVID-19 [20] report a higher prevalence of DM, up to 35%, possibly reflecting the increased severity of, rather than susceptibility to, COVID-19 in patients with DM. However, when analyzing these data, it is important to consider that the real prevalence of DM among COVID-19 patients might actually be higher than estimated, since most studies do not clarify whether DM diagnosis was self-reported or based on specific criteria, the percentage of undiagnosed individuals with DM is relatively high [34] and, furthermore, a number of patients with DM did not undergo COVID-19 testing to confirm infection due to self-isolation. All these factors may skew study results and thus underestimate the potential of DM to confer increased susceptibility to COVID-19.

In the other direction, SARS-CoV2 per se may affect glycemic control. It is well known that any acute illness or inflammatory condition may increase insulin resistance and, therefore, blood glucose levels [35]. Of note, among 1122 hospitalized patients with COVID-19 in 88 USA hospitals, approximately 40% had DM or uncontrolled hyperglycemia on admission [21]. Lessons from previous coronaviruses indicate that they can directly affect glucose metabolism since transient hyperglycemia was commonly seen in patients infected with SARS-CoV2 in 2003, even in those with mild disease [36]. It was later recognized that the binding of SARS-CoV2 to its receptor, angiotensin converting enzyme 2 (ACE2), in the endocrine section of the pancreas, may directly damage pancreatic islets and reduce insulin secretion, leading to transient insulin-dependent DM [22,23,37]. A similar hypothesis can be postulated for SARS-CoV2, which shares the same membrane receptor for entry into host cells with SARS-CoV2, namely ACE2 [38]. Interestingly, ACE2 expression is slightly higher in the pancreas compared to the lungs, indicating that SARS-CoV2 may also bind to pancreatic ACE2 and cause subclinical pancreatic injury [24]. A previous study reported enhanced ACE2 activity levels in the pancreas of diabetic mice [39]. Further studies are needed to verify whether clinically relevant damage of the endocrine pancreas can be observed in patients with severe COVID-19.

In autoimmune diseases, viruses may activate the immune system and cause increased synthesis and release of cytokines in genetically predisposed subjects. It has been intriguingly hypothesized that SARS-CoV2 may trigger such mechanisms [40]. Additional studies are needed to examine whether SARS-CoV2 can trigger the autoimmune pathophysiology of pancreatic β-cell destruction and lead to autoimmune DM in the long term in predisposed individuals.

### 3.2. Clinical Course of SARS-CoV2 Infection in Patients with DM

DM is associated with an increased incidence and severity of respiratory tract infections [41], and hyperglycemia on admission has been identified as a poor prognostic marker for pneumonia outcomes [42]. A known history of DM and elevated plasma glucose levels were independent predictors of mortality in patients with SARS [36]. Similarly, DM was found to be the comorbid condition with the strongest association with adverse outcomes in patients with Middle East respiratory syndrome (MERS) [43]. Regarding influenza, among hospitalized patients, the risk of ICU admission was found to be higher in patients with DM compared with those without [44].

Although not all data are consistent, COVID-19 seems to be more severe and fatal in patients with DM [18,21,45,46,47,48]. In a Chinese CDC report, patients with DM had the second highest fatality rate (7.3%) after cardiovascular disease among those with comorbid conditions [45]. Data from the first USA report regarding glycemic control in hospitalized patients with COVID-19 have shown that death rates were more than four times higher among people with DM or hyperglycemia than those without either condition [21]. The strongest scientific evidence comes from meta-analyses. In a meta-analysis of nine studies with a total of 1936 COVID-19 patients, the presence of DM was associated with increased severity of infection [46]. In a second meta-analysis of 30 studies including 6452 patients, the presence of DM was associated with increased mortality, severity and ARDS, while no difference was found for ICU admission [47]. In an observational study in 1317 patients with DM hospitalized for COVID-19 in 53 French centers, which explored the clinical and biochemical characteristics of DM patients associated with poor prognosis, an elevated body mass index (BMI), even in the overweight range, displayed a strong independent adverse association with the primary outcome, which was mechanical ventilation and/or death within the first week. In the same report, dyspnea and lymphopenia on admission, as well as elevated C-reactive protein and aspartate aminotransferase levels, were independent predictors of adverse outcomes, while advanced age and micro- and macrovascular complications of DM were independently associated with increased mortality [48]. In another retrospective study from Wuhan with a similar rationale, advanced age (>70 years) and comorbid hypertension were identified as independent risk factors predicting in-hospital mortality among patients with DM [49].

In contrast with the epidemiological data presented above, an observational study from 169 hospitals in Asia, Europe and North America including 8910 patients with COVID-19, reported that, although non-survivors had a greater prevalence of DM, the presence of DM was not an independent predictor of in-hospital death, whereas age, coronary heart disease, heart failure, chronic obstructive pulmonary disease and smoking were [50]. In the same direction, DM was not independently associated with in-hospital mortality in patients with COVID-19 according to a Chinese study [49]. In line with these findings, a meta-analysis of four studies showed that the severity of COVID-19 was not increased in patients with DM, but only in patients with hypertension and cardiovascular disease [18]. Nevertheless, it is noteworthy that the heterogeneity of the data in this meta-analysis was rather high and a large variation between the sample sizes of the included studies existed.

### 3.3. Mechanisms for the Increased Risk and Severity Associated with COVID-19 in DM

Infections represent a major challenge for patients with DM, who are more prone to develop severe, complicated infections compared with the general population [51]. The precise mechanisms linking DM with infections are not fully understood. The underlying pathophysiology is complex and multifactorial, but hyperglycemia seems to be the common denominator [52]. The increased vulnerability to infections is mainly related to a compromised immune function, but also to micro- and macrovascular complications of DM [53]. Furthermore, infection risk depends to a great extent on glycemic control [52]. Although several randomized clinical trials have shown that intensive glycemic control may reduce the risk of microvascular complications, the effect of intensive glycemic control on infection risk has not been adequately studied and observational studies report conflicting results [52].

Acute and chronic hyperglycemia may significantly affect both the innate and the adaptive immune system [54,55]. More specifically, hyperglycemia has been associated with an impaired neutrophil and macrophage function, comprising decreased neutrophil chemotaxis, adherence, phagocytosis and superoxide formation, leading to diminished intracellular killing capacity [55]. In addition, impaired T-cell-mediated immune response and ineffective microbial clearance have been described in patients with DM [56]. Interestingly, a recent study in COVID-19 patients showed that hypertension and DM resulted in the delayed clearance of SARS-CoV2 [57]. With regard to the respiratory system, hyperglycemia and insulin resistance may collectively damage surfactant D-mediated host defense in the diabetic lung [58]. Furthermore, loose junctions between airway epithelial cells, increasing the trans-epithelial glucose gradient and elevated glucose levels in the airway surface liquid due to hyperglycemia, may reduce the airway defense capacity against infection, leading to lung bacterial overgrowth in patients with DM [59].

ACE2 has been identified as the major receptor mediating SARS-CoV2 entry into host cells [38]. At the same time, it has been suggested that through angiotensin II catabolism, ACE2 has beneficial anti-inflammatory properties and may protect against severe lung injury [60]. Data regarding the expression of ACE2 in the lungs of patients with DM are scarce. An increased pulmonary ACE2 expression in DM, as shown by a Mendelian randomization analysis [61] and either due to DM per se or as a result of pharmacological treatment [62], may increase the susceptibility of patients with DM to SARS-CoV2 infection by facilitating the entry of the virus into the host. On the other hand, experimental data suggest that ACE2 expression is reduced in late-stage DM [39] and SARS-CoV2 infection may also downregulate ACE2 expression, predisposing patients to severe complications and adverse outcomes [63]. Further research is therefore needed to clarify the protective or detrimental role of ACE2 alterations in DM in terms of COVID-19 prognosis.

Patients with DM are characterized by a chronic low-grade inflammation associated with underlying insulin resistance and visceral fat accumulation [64]. Upon SARS-CoV2 infection, this pre-existing subclinical inflammation can be exacerbated, predisposing DM patients to the so-called “hyperinflammatory or cytokine storm syndrome”, which is the main driver of progression to severe life-threatening disease. Moreover, DM is characterized by a high risk of vascular thrombotic events [65]. Both acute and chronic hyperglycemia, combined with hyperinsulinemia and endothelial dysfunction, are associated with pro-thrombotic effects [66]. In parallel, COVID-19 is associated with coagulation abnormalities, particularly in patients with severe disease [67]. Therefore, the concomitant presence of two hyper-coagulation states may increase the risk of thrombotic complications. In addition, older age, cardiovascular and renal disease, hypertension and severe obesity are common comorbidities in people with DM and are also important risk factors for adverse COVID-19 outcomes [19,68]. Last, but not least, the milder and more atypical initial presentation of COVID-19 symptoms in patients with DM (the absence of classical signs and symptoms), may lead to delayed healthcare delivery and hence poorer prognosis.

### 3.4. Role of Glycemic Control for COVID-19 Outcomes

Glucose is an important fuel for white blood cells (WBCs); these cells use large amounts of glucose even under resting conditions [69]. During the activation of the immune function, a prominent feature of WBCs is to further increase glucose utilization, in order to meet the increased energy demands and acquire intermediates for macromolecule biosynthesis [70]. Therefore, the availability of glucose in the circulation during infection is of crucial importance; this is achieved by the development of insulin resistance, an evolutionarily preserved response that allows the host to survive [35]. Infection activates immune cells to secrete pro-inflammatory cytokines, which mediate: (a) insulin resistance in insulin-sensitive tissues by inhibiting insulin signal transduction and (b) the increased secretion of cortisol, epinephrine and norepinephrine, which in turn aggravate and sustain insulin resistance. Increases in glycogenolysis, anaerobic glycolysis and proteolysis in muscle and lipolysis in adipose tissue increase the production of substrates for gluconeogenesis (lactate, glycerol and amino acids, mostly alanine) and non-esterified fatty acids (NEFAs, potent stimulators of gluconeogenesis), leading to an increase in hepatic glucose production and blood glucose levels (stress hyperglycemia) [71]. These changes are accompanied by a balanced and tightly coordinated increase in insulin secretion, which helps to keep the metabolic system under control (Figure 1). The rationale behind the role of insulin resistance during infection has been provided by in vitro experiments showing that the activation of WBCs leads to the increased expression of glucose transporters (GLUT1, GLUT3 and GLUT4 isoforms) on the plasma membrane; the expression of GLUT3 and GLUT4 and glucose transport are further augmented by increases in insulin levels within the physiological range [72]. These mechanisms help to redistribute glucose as a source of energy away from peripheral tissues and direct it towards cells that mediate the immune response and are therefore crucial to survival. These data suggest that mild to moderate hyperglycemia is beneficial at times of infection, to provide fuel for the immune system, but also for the central and peripheral nervous system; in contrast, excessive hyperglycemia or hypoglycemia can be catastrophic since they both weaken the immune system and induce a number of serious and life-threatening complications.

Although DM has been associated with adverse outcomes in the COVID-19 pandemic [73], the available evidence is still limited. It has been suggested that DM and also excessive hyperglycemia on admission among individuals without a prior diagnosis of DM are strong predictors of poor outcomes, including death, in hospitalized patients with COVID-19 [9]. It should be noted that ~30% of individuals with type 2 DM (T2DM) are unaware of their condition and remain undiagnosed; this proportion may be even higher among those hospitalized with acute illnesses such as pneumonia [74]. In another study from Hubei Province in China, including 7337 hospitalized patients with COVID-19, preexisting T2DM was an important risk factor for poor outcomes and higher mortality rates [75]. In this study, patients with well-controlled T2DM (blood glucose levels 3.9–10 mmol/L, HbA1c 7.3%; 56.3 mmol/mol) had a lower need for medical interventions during hospitalization, fewer organ injuries and complications and a lower rate of fatal outcomes compared to those with poor metabolic control (HbA1c 8.1%) [75]. In contrast with these studies, a report from France, with a total of 1317 COVID-19 patients with DM and HbA1c ranging from <7% (<53 mmol/mol) to >9% (74.9 mmol/mol), found no correlation between the severity of glycemic control and tracheal intubation and/or death within 7 days of admission [48]. However, these results are difficult to interpret since HbA1c was available only in 64% of patients and was either determined on admission or retrieved from routine medical records from the previous 6 months.

The major physiologic function of insulin is to maintain anabolism, which requires blood glucose fluctuations within a very tight range (3.9–7.8 mmol/L); below or above this range, metabolism gradually enters a catabolic state [76]. It is therefore not surprising that either hypoglycemia (<3.9 mmol/L) and hyperglycemia (>11.1 mmol/L) in hospitalized patients have been associated with poor prognosis and adverse clinical outcomes, aggravating an already catabolic condition [77]. In support of this, studies looking at inpatient management of hyperglycemia, particularly in the ICU setting, suggest that blood glucose levels should be kept between 7.8–10 mmol/L in order to avoid excessive hyperglycemia or moderate/severe hypoglycemia and hinder multi-organ failure and fatal outcomes [78]. A large number of studies in patients with or without DM have shown a linear positive correlation between the degree of hyperglycemia upon admission and during hospitalization for critical illness and the risk of serious adverse events, including death [31].

Based on the above, good glycemic control prior to and during infection with COVID-19 may be extremely important for people with DM for several reasons:(i)Both hyperglycemia and hypoglycemia may disrupt an already malfunctioning innate immune system in patients with DM, increasing their susceptibility to infections [70]. Studies in poorly controlled subjects with T2DM have shown attenuated increases in plasma levels of cytokines and adhesion molecules after the administration of endotoxin in vivo [79]. These defects can be partly restored after adequate glycemic control, highlighting the importance of optimal glucose management for the maintenance of appropriate immune function in DM during infections [80].(ii)T2DM is an inflammatory disease characterized by the increased production of proinflammatory cytokines from adipose tissue/endothelial cells and insulin resistance in the liver/muscle/adipose tissue, along with inadequate β-cell function; by the time of clinical diagnosis, at least 50% of β-cell mass has been lost due to progressive destruction by oxidative stress, generated from the elevated levels of glucose and NEFAs in the circulation (glucotoxicity and lipotoxicity) [81]. In type 1 DM (T1DM), although it results from the primary loss of β-cell mass due to autoimmune processes with consecutive insulin deficiency, residual β-cell function may be retained in some individuals for years after clinical diagnosis; it should be noted that insulin resistance and an inflammatory state are also features of this type of DM [82,83,84]. In the presence of insulin resistance, poor glycemic control reflects relative insulin insufficiency for the corresponding metabolic needs; insulin resistance and the shortage of insulin may promote a catabolic state. In the course of COVID-19 infection, insulin requirements increase dramatically due to the “cytokine storm” and high insulin resistance [27]. Considering the fact that SARS-CoV2 may also directly attack the islets, severe insulin deficiency aggravates the catabolic state, generating massive increases in circulating glucose and NEFAs; these will lead to severe ketoacidosis or a hyperglycemic hyperosmolar state and multi-organ failure [27,85] (Figure 1).(iii)Hyperglycemia can be toxic for cells that take up glucose passively and independently of insulin, such as those of the central/peripheral nervous system, hepatocytes, endothelial/epithelial cells and pancreatic β-cells. Glucose overload increases oxidative stress, advanced glycation end-product formation and apoptosis, leading to cellular damage and severe clinical complications during critical illness. Glucose toxicity has been linked to the development of liver/kidney dysfunction, neuropathy, endothelial damage, susceptibility to bacterial infections and respiratory tract dysfunction [86]. Regarding COVID-19, an important issue to consider is the glucose overload of airway surface epithelial cells. Low glucose concentrations in the airway surface are an important part of normal lung defense against infections; high concentrations of glucose in airway epithelial cells could therefore predispose to bacterial growth and pulmonary infections [87]. Protection from neuropathy during critical illness also has important clinical implications, such as a shorter duration of mechanical ventilation and ICU hospitalization [86]. In a study in surgical ICU patients with and without DM, intensive glycemic control with insulin (targeting blood glucose levels between 4.4–6.1 mmol/L) reduced hospital-acquired infections and lethal sepsis, neuropathy, acute renal failure, blood transfusions and the risk of multiple organ failure and death versus conventional insulin treatment (blood glucose levels 10–11.1 mmol/L); these beneficial effects were attributed to the decrease in hyperglycemia and not to insulin use [88]. Similar results were reported in medical ICU patients with DM, including those with respiratory infections and septic shock, under intensive or conventional glycemic control with insulin: the lowest mortality occurred among patients with blood glucose levels between 3.9–5.5 mmol/L and increased consistently with blood glucose levels above this range, with the highest mortality at blood glucose levels ≥ 10 mmol/L [89]. Although these studies provided convincing evidence to support the significance of good glycemic control during critical illness, they both reported frequent hypoglycemic episodes in the group assigned to intensive glucose management. The question of which is the safest range of blood glucose levels for “benefit without harm” was finally answered by the NICE-SUGAR trial in medical ICU patients on intensive (blood glucose levels 4.5–6 mmol/L) vs. conventional glycemic control (blood glucose levels 8–10 mmol/L) by intravenous insulin: mortality in the latter group was actually lower than in the former group; the reason was more frequent episodes of moderate or severe hypoglycemia in the intensive treatment group, which was associated strongly with the risk of death [90]. Therefore, a target range of moderate hyperglycemia between 8–10 mmol/L is both effective and safe to reduce glucose toxicity, hypoglycemia risk and mortality during critical illness.(iv)Severe hyperglycemia may promote osmotic effects on cells, fluid shifts and electrolyte disturbances which, along with the diarrhea that occasionally accompanies COVID-19 infection, can lead to severe dehydration, decreases in tissue blood flow and ischemia, severe defects in cellular metabolism and ketosis [86,88]. It should be noted that in T2DM, endothelial dysfunction and impaired insulin-stimulated blood flow in major tissues, like adipose tissue and skeletal muscle, are already present early during the course of the disease and long before clinical diagnosis, playing a significant role in the pathophysiology of metabolic dysregulation [91,92]. Endothelial dysfunction is further aggravated in people with DM because SARS-CoV2 infection induces endotheliitis in several organs as a direct consequence of viral involvement and of the host inflammatory response [54].(v)Hypoglycemia (blood glucose levels < 3.9 mmol/L) is the most frequent acute complication of DM and the main obstacle to achieving optimal glycemic control; it is associated mostly with the use of insulin and sulfonylureas [93]. In patients with T1DM, hypoglycemia is very common and unpredictable; it can be severe when associated with unawareness of the preliminary warning symptoms, predisposing patients to life-threatening complications [94]. In addition to weakening the body’s defensive mechanisms to infections by depriving white blood cells and the brain of their main fuel [70,72], hypoglycemia can also impair autonomic function, cause vasoconstriction and ischemia, prolong the QT interval, predisposing to fatal cardiac arrhythmias especially at night during sleep, and has been associated with fatal outcomes in critically ill patients [95].

Taken together, the existing evidence suggests that uncontrolled hyperglycemia in DM may be associated with increased COVID-19 susceptibility and severity/mortality and could be an important risk factor for adverse clinical outcomes and poor prognosis, especially in the ICU setting. In poorly controlled DM, infection-related hyperglycemia is exacerbated, leading to aggressive insulin treatment, significant blood glucose fluctuations and potentially to ketosis; hyperglycemic emergencies can therefore be the initial clinical presentation of COVID-19 in these patients. Therefore, optimal metabolic regulation may be desirable for people with DM both prior to and after COVID-19 infection, to minimize the risk of severe clinical outcomes; however, in contrast to other infections, data for SARS-CoV2 are still limited.

In support of these suggestions, a recent study by Sardu et al., in patients with diabetes during COVID-19 infection (age ~67 years, BMI ~28 Kg/m^2^), showed that hyperglycemia (~11 mmol/L) displayed a higher cumulative incidence of severe disease: (a) both upon admission and during hospitalization for 18 days, hyperglycemia was positively associated with higher plasma IL-6 and D-dimer levels, indicating established inflammatory and coagulation abnormalities; (b) the administration of insulin and the correction of hyperglycemia decreased plasma IL-6 and D-dimer levels, decreasing the risk of severe outcomes (including disseminated intravascular coagulation) and death [96]. A clinically important observation in this study was that, in patients with hyperglycemia, for every 0.56 mmol/L drop in plasma glucose level between admission and 18 days of hospitalization, there was an 11% relative decrease in severe disease risk; however, the authors recognize the importance of avoiding the risk of hypoglycaemia during glycemic control with insulin, according to earlier suggestions [96]. Therefore, in line with this and previous studies, optimal glycemic control during COVID-19 infection may be protective against adverse events and should include the proper regulation of hyperglycemia (HbA1c < 7%; 53 mmol/mol), the avoidance of hypoglycemia and keeping blood glucose levels within a physiological range between 4.4–10 mmol/L during hospitalization and treatment with insulin [78].

## 4. General Recommendations for the Prevention of COVID-19 in Patients with DM

Similar to preventing other infectious diseases, the prevention of COVID-19 requires that patients with DM pursue a healthy lifestyle, abstain from smoking and comply strictly with medical recommendations. It is essential that the medical community sensitizes all patients with DM to the importance of maintaining optimal glycemic control in the period of the COVID-19 pandemic, since glycemic stability within the euglycemic range can help patients with DM have a milder clinical course of COVID-19 in case of infection, as already presented [97]. Optimal glycemic control can be accomplished with regular glucose monitoring and the appropriate adjustment of antidiabetic treatment according to medical instructions. General public health measures, such as social distancing, hand hygiene, face masks and home confinement, should be particularly emphasized in all patients with DM [27]. Maximal self-containment should be maintained with the use of telemedicine services [27]. In particular, connected health models and telemedicine applications should be implemented to encourage regular reviews and self-management education programs virtually, and ensure that patients are adherent to therapy [27,29]. Telephone advice, telemedicine consultations and the online coordination of drug delivery should be applied to reduce the exposure of DM patients, while the ensuring uninterrupted continuity of care. Routine non-emergency clinic visits and overcrowding in hospitals should be avoided to reduce disease spread among people with DM [27,29]. Stress management techniques, the control of DM-related comorbidities and routine vaccinations for pneumococcal pneumonia and seasonal influenza are important [29]. Patients should be advised not to prematurely discontinue any established antidiabetic or other treatment [27]. An increased vigilance is warranted for an early detection of even atypical signs and symptoms of SARS-CoV2 infection in patients with DM, and a lower clinical threshold for COVID-19 suspicion should be set to avoid delayed healthcare delivery and adverse outcomes in these patients [3]. Patients with poor glycemic control should be promptly referred to hospitals with the first symptoms of infection to facilitate early diagnosis and treatment and prevent progression to more severe stages of the disease [27,29].

## 5. General Recommendations for the Management of Hyperglycemia in Patients with DM and COVID-19

Once the first symptoms—even mild—of COVID-19 emerge, patients with DM should promptly contact their treating diabetologist or any other healthcare professional, preferably by phone, email or video call to minimize physical contact, and urgently seek medical advice. The major general principles underlying the optimal management of patients with DM and COVID-19 include an individualized patient-tailored therapeutic approach, rigorous glucose monitoring, the careful consideration of drug interactions and the achievement of optimal glycemic control through balanced nutrition, adequate hydration and physical activity, psychological stability and the appropriate adjustment of antidiabetic treatment according to “sick-day rules”, in line with medical recommendations [3,27,29].

### 5.1. Special Considerations for T1DM

To date, there are fortunately no reports of any children or adolescents with T1DM and severe life-threatening COVID-19, in line with the general notion that COVID-19 severity is typically associated with advanced age [98]. It is important to highlight that patients with T1DM and COVID-19 should never discontinue their insulin treatment due to their high risk of uncontrolled hyperglycemia and diabetic ketoacidosis (DKA) upon infection. According to a preliminary report from the USA summarizing data from a small number of patients with T1DM and confirmed COVID-19 (n = 33), the most prevalent presenting manifestation was hyperglycemia, and nearly one third of T1DM patients experienced DKA [99]. Infected T1DM patients should be in regular contact with their treating physician, ensure an adequate fluid intake and closely monitor their blood glucose and ketone levels under continuous medical surveillance [27,29]. In case blood ketone levels are higher than 3 mmol/L, patients should be promptly referred to hospital for further management.

### 5.2. Special Considerations for T2DM

#### 5.2.1. Glycemic Goals

Glycemic goals should always be individualized according to patients’ age, comorbidities, complications and the clinical severity of infection [30]. For non-senile diabetic patients with mild infection, fasting glucose levels should be below 6.1 mmol/L and 2-h postprandial glucose levels below 7.8 mmol/L. For older patients with mild infection, fasting glucose levels should be below 7.8 and 2-h postprandial glucose levels below 10 mmol/L. For older patients hospitalized with severe SARS-CoV2 infection, glycemic goals are less stringent and fasting glucose levels up to 10 mmol/L or postprandial/random levels up to 13.9 mmol/L may be acceptable [100].

#### 5.2.2. Glucose Monitoring

In hospitalized T2DM patients with COVID-19, tight glucose monitoring is essential to optimize clinical outcomes [30,101]. Glucose monitoring should preferably be continuous subcutaneous (CGM) to minimize physical contact between patient and nurse and obtain a comprehensive picture of the daily glycemic profile [102]. In the hospital setting, CGM is still considered investigational, or at least it has not been recommended to replace the point-of-care glucose measurement and become a standard-of-care technology. Using CGM devices in the “traditional” way still carries a risk of viral transmission, as there is still direct contact between the patient and the physician or nurse. Remote monitoring with CGM devices, such as DEXCOM and Medtronic, represents a more efficient means of reducing exposure and risk of COVID-19 transmission. The main goals of in-hospital glucose monitoring are to limit abnormal glycemic variability (time in range >70%) and minimize hypoglycemic episodes (time in hypoglycemia <4%) [103]. Increased glycemic variability, defined as fluctuations between abnormally high and low glucose levels [104], has been recognized as an important adverse prognostic factor for critically ill patients, especially in the ICU setting [105]. Hypoglycemia can be also detrimental for severely infected patients, since it stimulates the release of stress hormones, such as cortisol and catecholamines, leading to elevated levels of circulating inflammatory mediators and the augmentation of pre-existing inflammation [106], and may also exert arrhythmogenic effects by prolonging the QT interval, thus increasing cardiovascular mortality [94]. In this context, frequent episodes of hypoglycemia and/or cardiac autonomic neuropathy might act synergistically with specific anti-COVID-19 drugs, such as azithromycin, in terms of QT prolongation and predispose to potentially life-threatening ventricular arrhythmias [107,108].

#### 5.2.3. Pharmacological Treatment

Preclinical data have reported an association of several antidiabetic medications with an upregulated ACE2 expression, raising concerns about a possible risk of facilitating SARS-CoV2 infection in T2DM patients under these treatments [62]. These drug classes include insulin, thiazolidinediones (TZDs), sodium glucose co-transporter 2 (SGLT2) inhibitors and glucagon-like peptide 1 receptor (GLP-1R) agonists. However, this is just a speculation based on animal and in vitro data, as evidence in humans is lacking, and the benefits of these drugs by far outweigh any theoretical risks. The following general recommendations can be made for the major antidiabetic drug classes (Table 1):

Insulin: Insulin should be continued, and the insulin dose should be adapted depending on glycemic control, hypoglycemia risk, the severity of infection and concomitant drug treatment. Serum potassium balance should be monitored in all insulin-treated patients with COVID-19, since SARS-CoV2 infection may reduce serum potassium levels and exacerbate insulin-induced hypokalemia, by downregulating the expression of ACE2, thus leading to increased circulating levels of angiotensin II and subsequent hyperaldosteronism, which may result in increased renal potassium wasting [27]. Intravenous insulin infusion has proven to be an effective option to achieve glycemic targets and improve outcomes in hospitalized patients with T2DM and COVID-19 [96]. According to clinical experience, insulin requirements can be very high in hospitalized patients with severe hyperglycemia and/or DKA.

Metformin: Metformin should be discontinued in severe hospitalized cases due to the risk of lactic acidosis (a rare but potentially lethal complication). Renal function should be closely monitored [27].

Sulfonylureas: Sulfonylureas should be discontinued if regular oral food intake cannot be maintained due to the high risk of hypoglycemia. The hypoglycemia risk is greater with the concomitant use of anti-COVID-19 treatments, such as chloroquine and hydroxychloroquine [29].

SGLT2 inhibitors: There is no evidence of safety or efficacy for use in the hospital setting. SGLT2 inhibitors should be discontinued in severe hospitalized cases due to the risk of euglycemic DKA precipitated by dehydration (intravascular volume depletion) and relative insulinopenia [27,29]. Their use in patients with severe infection is not appropriate. They could be further continued in mild outpatient cases due to their significant cardio- and renoprotective effects [109]. SGLT2 inhibitors have been shown to reduce mortality, major adverse cardiovascular events and heart failure hospitalization in T2DM patients at high cardiovascular risk [110,111,112]. They can also attenuate the slope of renal function decline in T2DM patients with diabetic nephropathy [113]. It needs to be emphasized that, for T2DM patients at risk of or with established COVID-19, the preservation of optimal cardiovascular and renal function is critical for achieving favorable clinical outcomes. In this context, a multicenter phase III randomized clinical trial has recently been launched (DARE-19 Study: Dapagliflozin in Respiratory Failure in Patients With COVID-19, ClinicalTrials.gov Identifier: NCT04350593), aiming to assess the safety and efficacy of dapagliflozin vs. placebo in reducing disease progression, complications and mortality in hospitalized patients with mild–moderate COVID-19 with risk factors for developing severe complications.

Dipeptidyl peptidase 4 (DPP-4) inhibitors: DPP-4 inhibitors should be further continued in mild outpatient cases [27,29]. Recently, human DPP-4, the incretin-degrading enzyme that is up-regulated in patients with T2DM, obesity and metabolic syndrome [114,115], has been proposed as a potential therapeutic target for COVID-19 [116], given that DPP-4 was identified as the functional receptor for the spike protein of MERS-CoV, facilitating its entry into host cells [117]. It was also suggested that the higher rate of mortality and complications in patients with T2DM and MERS-CoV infection could be partly associated with a DPP4-mediated dysregulated immune response since DPP-4 is expressed in immune cells among other tissues and promotes immune cell activation [118]. Further preclinical observations have fueled an intriguing debate about whether DPP-4 inhibitors, which are commonly prescribed in patients with T2DM, might protect against SARS-CoV2 infection and mitigate COVID-19 severity [116,119,120,121]. It has been shown in vitro that antibodies targeted against DPP-4 can prevent the infection of human bronchial epithelial cells by the human Coronavirus-Erasmus Medical Center (hCoV-EMC) [117]. It has also been shown that transgenic mice expressing human DPP-4 are more susceptible to MERS-CoV infection and develop an abnormal inflammatory response, leading to a lethal respiratory infection [122,123]. On the other hand, there are various arguments against the hypothesis that DPP-4 inhibitors may represent a novel therapeutic approach for COVID-19. First, the inhibition of hCoV-EMC infection could be demonstrated only with DPP-4 neutralizing antibodies and not with clinically applied DPP-4 inhibitors, such as sitagliptin, vildagliptin, saxagliptin, linagliptin and alogliptin [117], highlighting the need to make a distinction between DPP-4 inhibitors and DPP-4 inhibition. In addition, the possible interaction between SARS-CoV2 spike glycoproteins and DPP-4 has been predicted only by structural studies without solid confirmation in human cells [124]. The effects of DPP-4 inhibitors on immune function and infection risk are also controversial, since there have been reports of an increased risk of upper respiratory tract infections with these agents [125], however, this has not been confirmed in other studies [126]. Based on these limitations, it is rather premature to draw safe conclusions regarding the potential beneficial effects of DPP-4 inhibitors in the battle against COVID-19.

GLP-1R agonists: GLP-1R agonists should be continued with caution in outpatient cases (dehydration risk), and discontinued in severe hospitalized cases [27,29]. Considering that low-grade inflammation, excess adiposity, insulin resistance and compromised cardiovascular function are all associated with an unfavorable prognosis in COVID-19 [26], it is important to emphasize the anti-inflammatory, anti-adipogenic, insulin-sensitizing and cardioprotective effects of these drugs [127]. GLP-1R agonists have, interestingly, shown a strong anti-inflammatory potential by modulating macrophage infiltration and M1/M2 polarization via GLP-1 receptor signaling [128]. Of note, GLP-1R agonists are the second antidiabetic drug class, along with SGLT2 inhibitors, which have consistently shown a beneficial impact on cardiovascular and renal outcomes in T2DM patients at high cardiovascular risk [129].

TZDs (mainly pioglitazone): There is no evidence of safety or efficacy for use in the hospital setting. TZDs should be discontinued in severe hospitalized cases due to safety concerns about fluid retention and heart failure deterioration [29]. On the other hand, the significant insulin-sensitizing and anti-inflammatory effects of these agents [130,131], as well as their potential to ameliorate hepatic steatosis and inflammation [132], should not be disregarded.

Renin–angiotensin system (RAS) blockers: This antihypertensive drug class comprises angiotensin-converting enzyme inhibitors (ACEIs) and angiotensin II type 1 receptor blockers (ARBs), and is commonly prescribed in DM patients with hypertension, heart failure and/or diabetic nephropathy. There have been some initial concerns regarding the potential of these agents to predispose hypertensive patients to severe SARS-CoV2 infection due to their associated upregulated expression of membrane-bound ACE2 [133], which is the major receptor for the entry of SARS-CoV2 into host cells [38]. Based on their mechanism of action, ACEIs and ARBs can lead to elevated circulating levels of angiotensin I and II, which are the major substrates for ACE2. The increased substrate load may lead to a compensatory upregulation of ACE2 expression in several tissues, predominantly the heart, kidney and renal vasculature [134,135,136]. The upregulated ACE2 expression in response to RAS blockers has been mainly documented in the hearts of rats [135], and is more pronounced with ARBs compared to ACEIs [136]. Based on these preclinical experimental data, it was initially postulated that ACEIs and ARBs might carry an excess risk of adverse outcomes in COVID-19 patients, since an upregulated ACE2 expression in the respiratory epithelium may facilitate viral entry into host cells and increase susceptibility to severe infection [133]. However, it has been suggested that the potential of RAS blockers to lead to an upregulated ACE2 expression is dose-dependent and varies significantly with the type of ARB (compound-specific) and the tissue involved [63,137]. Of note, the expression of ACE2 in the lung is much lower than in other tissues. Furthermore, a recent study reported that neither ACEIs nor ARBs were found to be independent predictors of mortality in hospitalized patients with COVID-19 [50]. On the other hand, opposing data suggest that ACE2 might actually protect against severe respiratory infections, by converting angiotensin II into Ang II (1–7) which has important anti-inflammatory properties, so that ACEI expression, leading to increased ACE2 expression, may in fact be beneficial and positively modulate COVID-19 prognosis [60,138]. Based on these heterogeneous and contradictory data, the absence of high-quality scientific evidence and the lack of replication of preclinical findings in humans, there is currently no evidence-based recommendation to discontinue these otherwise beneficial drugs with organ-protective effects in patients with T2DM in order to prevent or mitigate the severity of SARS-CoV2 infection [63]. This position statement has been officially endorsed by numerous scientific societies, including the American Heart Association (AHA), the American College of Cardiology (ACC), the Heart Failure Society of America and the European Society of Cardiology [139,140].

Non-steroidal anti-inflammatory drugs (ibuprofen): Based on in vitro data showing an increased ACE2 expression with ibuprofen [133], it was originally suggested that ibuprofen might increase the risk of severe and potentially fatal COVID-19 and should therefore be avoided [141]. These initial safety concerns were based on anecdotal evidence originating from the French Health Minister and the WHO, but they were not further confirmed. In a retrospective cohort study from Israel in 403 confirmed cases of COVID-19, ibuprofen use was not associated with worse clinical outcomes compared to paracetamol or no antipyretics at all [142]. Taken together, the current epidemiological evidence is not strong enough to support a causal link between ibuprofen use and poor outcomes in COVID-19 [143]. Evidence from mechanistic studies alone should not be used to make strong statements against the use of any medications. All official statements emphasize the lack of solid evidence to recommend against the use of ibuprofen in patients with COVID-19. Due to its better safety profile, the WHO recommends using paracetamol monotherapy for fever reduction in patients with COVID-19, but ibuprofen can be considered if paracetamol fails to relieve patient symptoms [144].

#### 5.2.4. Impact of COVID-19 Treatment on Glucose Metabolism

Role of dexamethasone: Intravenous corticosteroids (CSs) are mainly indicated for mechanically ventilated COVID-19 patients with ARDS and are given for a minimal period of time to limit side effects [145]. Small retrospective cohort studies and case series evaluating short courses of CSs in patients with COVID-19 have reported both beneficial [146,147,148,149] and harmful effects [150,151]. Dexamethasone, the most potent CS, has recently emerged as a low-cost therapeutic intervention able to reduce mortality in hospitalized patients with severe COVID-19. These promising data were generated from a large, multicenter, open-label trial in the United Kingdom, which randomized patients either to dexamethasone or to standard-of-care treatment and assessed 28-day mortality as the primary endpoint (RECOVERY Study: Randomized Evaluation of COVID-19 Therapy) [152]. A preliminary analysis of this study in 6425 participants has shown that dexamethasone can reduce mortality by 35% in mechanically ventilated patients and by 20% in patients requiring supplemental oxygen but not mechanical ventilation at enrolment. No benefit was demonstrated for patients with milder disease who did not require respiratory support at randomization. Based on these preliminary data, the US COVID-19 Treatment Guidelines Panel recommends the administration of dexamethasone at a dose of 6 mg per day (orally or intravenously) for up to 10 days in all COVID-19 patients under mechanical ventilation, and also in patients requiring oxygen supplementation [153]. In dexamethasone-treated patients, clinicians need to closely monitor for CS-induced adverse effects, mainly hyperglycemia, the reactivation of latent infections and drug interactions. Hyperglycemia is the most common and clinically relevant side effect. CSs counteract the endogenous metabolic actions of insulin and exert hyperglycemic effects by reducing insulin secretion and primarily increasing peripheral insulin resistance [154]. Dexamethasone in particular has shown the potential to reduce insulin-mediated glucose transport and phosphorylation, as well as glycogen synthesis and glucose oxidation in rat skeletal muscle, by directly inhibiting the translocation of glucose transporter 4 (GLUT4) to the plasma membrane in response to insulin [155]. Given these potent insulin resistance-promoting effects, dexamethasone can either trigger the new onset of DM in predisposed individuals or significantly deteriorate glycemic control in previously diabetic patients. It is therefore mandatory to intensify insulin treatment in order to prevent and handle steroid-induced hyperglycemic excursions in the hospital setting.

Role of vitamin D: Low vitamin D levels have been associated with an increased inflammatory cytokine response, an elevated risk of acute viral respiratory infections and an increased propensity for thrombotic complications [156]. Of note, vitamin D deficiency is commonly observed in patients with obesity and DM who carry an excess risk for poor clinical outcomes upon SARS-CoV2 infection. The antiviral effects of vitamin D have been attributed both to a direct interference with viral replication and to immunomodulatory and anti-inflammatory effects [157,158]. Interestingly, regulatory T lymphocytes (Tregs) which are considered to be the major defense against viral infection and uncontrolled inflammation, can be increased with vitamin D supplementation [156]. In animal models of ARDS, vitamin D treatment can reduce lung permeability by modulating ACE2 expression and activity [159]. The role of vitamin D in preventing COVID-19 infection and progression is currently not clear. A negative correlation has been reported between vitamin D levels and the prevalence of COVID-19 in European countries [160]. However, evidence on the association between vitamin D status and COVID-19 severity or mortality remains insufficient, highlighting the need for further epidemiological studies in this field. Based on the available evidence, people at high risk of severe vitamin D deficiency during the COVID-19 pandemic (obese, diabetic, renal dysfunction, minimal sun exposure) should consider vitamin D supplementation to maintain circulating 25(OH)-D concentrations at optimal levels (75–125 nmol/L) [160]. At present, more than 10 clinical trials are registered in the clinical trial registry of the National Institutes of Health (NIH), testing the effects of vitamin D supplementation at several doses in patients with COVID-19 [161].

Role of immune therapies: Anti-interleukin-6 (IL6) treatment strategies (tocilizumab or Janus kinase inhibitors) might be particularly effective in T2DM patients with severe COVID-19. According to a recent study providing a detailed biochemical analysis of T2DM patients hospitalized with COVID-19, IL-6 is the cytokine which is more pronouncedly increased in the serum of T2DM patients with COVID-19, making thus IL6-targeted interventions a promising therapeutic approach for these patients [162].

## 6. Knowledge Gaps and Questions to Be Answered

Most available studies so far refer only to the presence or absence of DM [163]. However, little is known about the role of the type of diabetes (T1DM vs. T2DM), the duration of DM, the specific antidiabetic treatment regimens, the presence of diabetes-related cardiometabolic complications, the concomitant presence of total and visceral adiposity and the degree of underlying insulin resistance in terms of determining prognosis and predicting clinical outcomes both in mild and more severe stages of COVID-19. More data are needed to address any potential impacts (beneficial or harmful) of different antidiabetic drug categories on COVID-19 outcomes in patients with DM.

Reflecting the uncertainties in the field, the following questions remain unanswered and need to be addressed with well-designed studies in the near future (some already running):Are patients with DM more susceptible than the general population to contracting SARS-CoV2, or is their susceptibility mainly limited to the increased severity of infection?Is the association of DM with adverse COVID-19 outcomes independent of coexisting risk factors such as advanced age and cardiovascular and kidney disease?Which antidiabetic drugs, if any, could interfere with COVID-19 prognosis, by either positively or negatively modulating clinical outcomes?Are SGLT2 inhibitors adequately safe for DM patients with COVID-19? Are they associated with better survival and cardiovascular and renal protection?What are the optimal glycemic targets for optimizing outcomes in patients with mild and severe forms of COVID-19?What are the exact clinical and biochemical characteristics of patients with T2DM (age, obesity, glycemic control, T2DM-related complications, insulin resistance, subclinical inflammation) which may serve as prognostic markers and determine poor prognosis in COVID-19?Based on their pathophysiological milieu, are patients with DM expected to gain particular benefits from specific antiviral therapeutic approaches (immunomodulatory, cytokine-targeted or other)?What is the immune response of people with DM infected with SARS-CoV2? Do they develop protective antibodies against the virus?Will the vaccines under development be equally safe and effective in DM patients as in the general population?

## 7. Concluding Remarks

Patients with DM have experienced and continue to experience significant challenges during the period of the COVID-19 pandemic. Even if they are not infected, they are at risk of dysregulated glycemic control due to the overall restrictive measures which compromise and disrupt the quality of healthcare delivery, especially to patients with chronic diseases. Moreover, restrictive measures in many countries have restricted the physical activity of city dwellers. In addition, many people with diabetes discontinued scheduled visits for diabetes control either because hospitals discontinued regular outpatient clinics or because people with diabetes, due to the fear of SARS-CoV2 exposure in hospitals, canceled their scheduled visits. In the case of infection, patients with DM have an increased risk of developing severe and potentially fatal complications. Nevertheless, it would be inappropriate to conclude that all patients with DM have an equally increased risk for severity and mortality due to COVID-19. Certain clinical and biological characteristics determine high risk phenotypes within the DM population and such prognostic markers need to be clearly characterized in future studies. Whether these phenotypic features include long-standing DM, advanced age, concomitant obesity and other cardiometabolic complications, profound insulin resistance or subclinical inflammation remains to be determined. Further research is also needed to examine which subgroups of patients with DM are expected to benefit the most from specific antiviral, immunomodulatory and other treatment strategies in the context of patient-tailored precision medicine, which emerges as an urgent priority in the era of COVID-19.

## Figures and Tables

**Figure 1 jcm-09-02288-f001:**
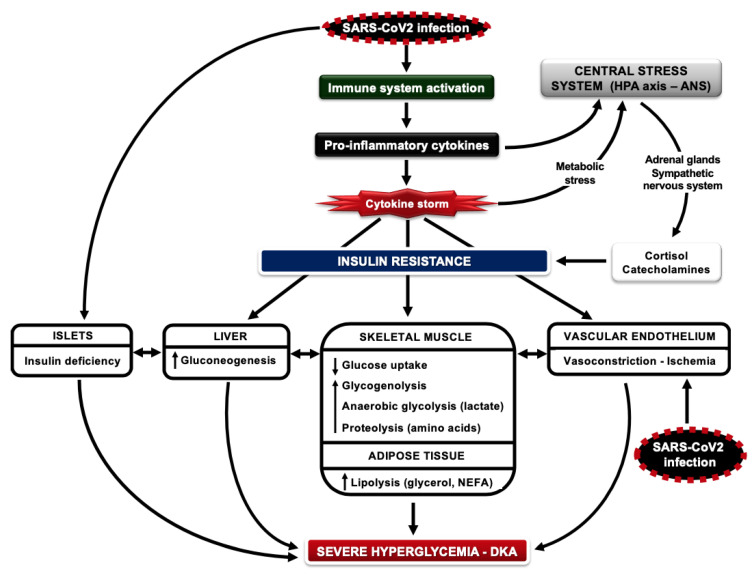
Severe acute respiratory distress syndrome coronavirus 2 (SARS-CoV2) infection activates the immune system, resulting in the secretion of pro-inflammatory cytokines, which mediate: (a) insulin resistance in insulin-sensitive tissues such as the liver, skeletal muscle, adipose tissue and vascular endothelium by inhibiting insulin signal transduction and (b) increased activation of the hypothalamic–pituitary–adrenal (HPA) axis and autonomic nervous system (ANS), enhancing the secretion of cortisol, epinephrine and norepinephrine, which in turn aggravates and sustains insulin resistance. Moreover, SARS-CoV2 infection induces endotheliitis in several organs as a direct consequence of viral involvement and of the host inflammatory response. Increases in glycogenolysis, anaerobic glycolysis and proteolysis in muscle and lipolysis in adipose tissue increase the production of substrates for gluconeogenesis and of non-esterified fatty acids (NEFAs), leading to an increase in hepatic glucose production and blood glucose levels (stress hyperglycemia). Endothelial dysfunction causes vasoconstriction, ischemia and coagulation abnormalities. SARS-CoV2 may directly damage pancreatic β-cells, reducing insulin release. The combination of hyperglycemia, insulin deficiency, the overutilization of NEFAs and dehydration can predispose to the development of diabetic ketoacidosis (DKA) in patients infected with SARS-CoV2. Overactivation of the immune system induces a strong inflammatory response, which can be enormous in critically ill patients (cytokine storm), further activating the HPA axis and ANS, enhancing insulin resistance and metabolic abnormalities.

**Table 1 jcm-09-02288-t001:** Clinical recommendations, special considerations and concerns for the use of specific antidiabetic drug categories in patients with T2DM and COVID-19.

Diabetes Pharmacotherapy	Clinical Recommendation	Special Considerations in the Setting of COVID-19
Insulin	Continue Adjust dose depending on glycemic control, hypoglycemia risk, severity of infection and concomitant drug treatment	Monitor serum potassium levels to prevent hypokalemia Insulin requirements can be very high in hospitalized patients with severe hyperglycemia and/or DKA
Metformin	Discontinue in severe hospitalized cases with hypoxia and hemodynamic instability due to the risk of lactic acidosis	Monitor renal function
Sulfonylureas	Discontinue if regular oral food intake cannot be maintained due to the risk of hypoglycemia	Hypoglycemia risk greater with the concomitant use of anti-COVID-19 treatments such as chloroquine and hydroxychloroquine
Sodium glucose co-transporter 2 (SGLT2) inhibitors (dapagliflozin, canagliflozin, empagliflozin)	Discontinue in severe hospitalized cases due to the risk of euglycemic DKA precipitated by dehydration and insulinopenia Continue in mild outpatient cases due to their significant cardio- and renoprotective effects	For T2DM patients at risk of or with established COVID-19, the preservation of optimal cardiovascular and renal function is critical for achieving favorable clinical outcomes It remains to be determined whether they can safely reduce complications and mortality in hospitalized patients with mild–moderate COVID-19 with risk factors for severe complications (DARE-19)
Dipeptidyl peptidase 4 (DPP-4) inhibitors (alogliptin, vildagliptin, sitagliptin, saxagliptin, linagliptin)	Continue in mild outpatient cases due to their good safety profile and possible use across a wide range of renal function	It is premature to draw safe conclusions regarding potential beneficial effects of DPP-4 inhibitors against COVID-19
Glucagon-like peptide 1 receptor (GLP-1R) agonists (liraglutide, dulaglutide, semaglutide, exenatide-extended release, lixisenatide, albiglutide)	Continue with caution in mild outpatient cases Discontinue in severe hospitalized cases	Consider possible dehydration risk due to gastrointestinal side effects (nausea/vomiting) Maintain adequate fluid intake and regular meals Anti-inflammatory, anti-adipogenic, insulin-sensitizing and cardioprotective effects relevant for COVID-19 prognosis
Thiazolidinediones (pioglitazone)	Discontinue in severe hospitalized cases due to safety concerns about fluid retention and heart failure deterioration	Keep in mind insulin-sensitizing and anti-inflammatory effects and their potential to ameliorate hepatic steatosis and inflammation

COVID-19: coronavirus disease 2019; DKA: diabetic ketoacidosis; T2DM: type 2 diabetes mellitus.

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
