# Peer review of "Clinical Management of Diabetes Mellitus in the Era of COVID-19: Practical Issues, Peculiarities and Concerns"

_jcm, 2020, doi:10.3390/jcm9072288_

Round 1

Reviewer 1 Report

This is an interesting and timeous review.

I am aware that the authors cannot cover all asects but:

  1. there is no discussion on how to manage patients with diabetes of how to continue seeing paptients during the era of COVID (telephone/skype etc)
  2. there is no mention of different ethnic groups and their susceptibility to infection and mortality risk - all deprivation-related??
  3. the test for COVID is poor with 20-25% false negative and a small number of false positive results
  4. what is the role of self-isolating and not being tested in skewing results from studies? Particularly nursing home deaths will have not had COVID tests.
  5. The English pre-published papers are interesting. Nineteen percent of patients with COVID hd diabetes suggesting increased susceptibilty. Smoking appeared protective!
  6. what is the role of dexamethasone?
  7. what is the role of CPAP vs ventilation?
  8. the experience in the UK suggests very high insulin rquirements in patients with hyperglycaemia/DKA.

There is a paragraph on hydroxychlorquine in this review. I believe that hydroxychlorquine has no role to play in COVID management.

Is there a role for VItamin D?

There is discussion on a number of drugs and COVID - what is the situation with COVID and ibuprofen?

There is no consistency with blood glucose levels - mg/dl need to be provided with mmol/L all through the text.

Author Response

Authors’ Reply to Reviewer #1 Comments

We thank the reviewer for the critical input and the helpful suggestions, which helped us revise and improve the quality of our manuscript. We considered all comments, performed all necessary revisions and we provide below a detailed point-by-point response to every single comment, hoping to have adequately addressed all concerns. All changes are denoted with red colour in the revised version of our manuscript.

  1. There is no discussion on how to manage patients with diabetes of how to continue seeing patients during the era of COVID (telephone/skype etc).

Authors’ response: In the revised version of our manuscript, we discuss in detail the need to implement telemedicine applications in the era of COVID-19 pandemic in order to promote self-management education programmes and ensure uninterrupted continuity of care for patients with DM. This new information is provided on page 14, section 4 (General recommendations for the prevention of COVID-19 in patients with DM), lines 428-433. Furthermore, we emphasize the need for patients with DM to contact their treating physician and seek medical advice by telephone, email or video call rather than direct personal contact on page 14, section 5 (General recommendations for the management of hyperglycemia in patients with DM and COVID-19), lines 444-446.

  1. There is no mention of different ethnic groups and their susceptibility to infection and mortality risk - all deprivation-related??

Authors’ response: There are indeed substantial racial and ethnic disparities in COVID-19 health outcomes. Social and health inequities predispose vulnerable populations to adverse clinical outcomes in pandemic situations, and this is especially true in the case of COVID-19 pandemic. COVID-19 crisis affects predominantly socioeconomically deprived populations, with Afroamerican black ethnic minorities being disproportionately affected in terms of COVID-19 incidence and mortality. We present these new data on page 2, Introduction, lines 47-53 in the revised version of our manuscript.  

  1. The test for COVID is poor with 20-25% false negative and a small number of false positive results.

Authors’ response: The reviewer is correct that the accuracy of molecular diagnostic tests for COVID-19 is suboptimal. PCR tests are highly specific (false-positive results only 5%) but their sensitivity is moderate and varies depending on several factors, primarily the technical quality of sample collection. The reported false-negative rates of COVID-19 diagnostic tests range between 2 and 29%, yielding a sensitivity of 71-98%. A single negative test cannot rule out infection in case of strong clinical suspicion. In the revised manuscript, these data are presented on page 2, Introduction, lines 53-61.  

  1. What is the role of self-isolating and not being tested in skewing results from studies? Particularly nursing home deaths will have not had COVID tests.

Authors’ response: The fact that a number of patients with DM will not undergo COVID-19 testing to confirm infection due to quarantine and self-isolation may indeed skew study results and underestimate the coexistence of DM with COVID-19. It is also true that deaths in nursing home residents with DM due to COVID-19 can often be under-assessed or under-reported, because no tests are usually performed in these populations to confirm the causal relationship of death with COVID-19. We refer to these limitations on page 5, section 3 (The association between DM and COVID-19), subsection 3.1 (Causality), lines 135-137. These limitations can partly explain the significant heterogeneity of data regarding the prevalence of DM in patients with COVID-19, or inversely the prevalence of COVID-19 in patients with DM, highlighting the difficulty to conclude to what extent does the presence of DM increase susceptibility to COVID-19.      

  1. The English pre-published papers are interesting. Nineteen percent of patients with COVID had diabetes suggesting increased susceptibility. Smoking appeared protective!

Authors’ response: Thank you for this comment. The prevalence of DM in patients with confirmed COVID-19 (mainly hospitalized) varies broadly between studies and depends to a great extent on methodological characteristics and/or limitations of each study. Paragraph 1 of section 3 (The association between DM and COVID-19), subsection 3.1 (Causality), on page 5 of the revised manuscript, is dedicated to addressing the issue of susceptibility of DM patients to SARS-CoV2 infection, and summarizes the major epidemiological evidence in this field. As critically discussed in this section, heterogeneity of data, poor reporting and the effect of self-isolation leading to not being tested for COVID-19 (as correctly suggested by the reviewer in Comment 4), may all skew study results and provide variable data on DM prevalence in COVID-19 patients, making it hard to conclude to what extent does DM independently increase susceptibility to COVID-19.

With regard to smoking, there have been indeed some reports from China, Italy, Spain and USA, showing a lower prevalence of smoking among hospitalized patients with COVID-19 compared to the general population. This lower smoking prevalence has been provocatively perceived as evidence for putative protective effects of smoking in COVID-19. However, it is rather unlikely that smoking confers any kind of protection against COVID-19, since the lower prevalence of smoking in COVID-19 patients may be in fact the result of biases, misreporting and unidentified confounding factors rather than reflect a genuine protective effect. In support of this hypothesis, a recent meta-analysis of 19 peer-reviewed papers from China, Korea and USA, addressing the association between smoking behaviour and COVID-19 severity, has shown that smoking may nearly double the rate of COVID-19 progression (Patanavanich R, Glantz SA. Smoking Is Associated With COVID-19 Progression: A Meta-analysis.  Nicotine Tob Res. 2020; ntaa082. doi: 10.1093/ntr/ntaa082). Based on these data and considering the multiple well-documented detrimental effects of smoking on lung function and immunity, global health authorities acknowledge smoking as a risk factor for COVID-19 and recommend smoking cessation. We decided on purpose not to refer to the smoking controversy in our paper to avoid the risk of conveying potentially misleading messages. We emphasize thus in our paper that all patients with DM should abstain from smoking in the context of pursuing a healthy lifestyle in order to prevent COVID-19 infection (page 14, section 4, line 419).    

  1. What is the role of dexamethasone?

Authors’ response: At the time of submitting the original version of our manuscript, data on dexamethasone in patients with COVID-19 (RECOVERY study) were not available. In response to the reviewer’s comment, we now provide detailed information on the role of dexamethasone as a low-cost therapeutic intervention able to reduce mortality in hospitalized patients with severe COVID-19 requiring mechanical ventilation or oxygen supplementation, with particular focus on hyperglycemia as a common and clinically relevant side effect.  Given its potent effects in promoting peripheral insulin resistance, dexamethasone can either trigger the new onset of DM in predisposed individuals or significantly deteriorate glycemic control in previously diabetic patients. The novel section on the role of dexamethasone for patients with COVID-19 can be found on pages 22-23, subsection 5.2.4 (Impact of COVID-19 treatment on glucose metabolism), lines 625-661.

  1. What is the role of CPAP vs ventilation?

Authors’ response: Patients with severe respiratory failure due to COVID-19 should be intubated and mechanically ventilated early in the disease course to slow disease progression and improve prognosis. However, in the setting of overwhelmed ICUs and shortage of ventilators, alternative non-invasive ventilatory support options have been tested, including continuous positive airway pressure (CPAP) and high-flow nasal cannula (HFNC). These devices can temporarily alleviate respiratory failure, but they have been associated with an increased risk of nosocomial viral transmission due to virus aerosolization. These new data on treatment of respiratory failure in patients with COVID-19 are now presented on pages 2-3, Introduction, lines 61-70, in the revised version of our manuscript.

  1. The experience in the UK suggests very high insulin requirements in patients with hyperglycaemia/DKA.

Authors’ response: It is true that according to clinical experience, insulin requirements can be very high in hospitalized patients with severe hyperglycemia and/or DKA. We make this statement on page 19, section 5.2.3 (Pharmacological treatment), subsection Insulin, lines 517-519, as well as on page 17, Table 1, as a special consideration for insulin use in the setting of COVID-19, as requested by the reviewer. 

  1. There is a paragraph on hydroxychloroquine in this review. I believe that hydroxychloroquine has no role to play in COVID management.

Authors’ response: At the time of original submission, hydroxychloroquine was one of the most studied treatment options for COVID-19. In the meantime, its role has been diminished on the basis of novel evidence. In the revised manuscript, we deleted the section on (hydroxy)chloroquine and its impact on glucose metabolism. Instead, in section 5.2.4 (Impact of COVID-19 treatment on glucose metabolism), we present the role of dexamethasone, Vitamin D and immune therapies (pages 22-24). 

  1. Is there a role for Vitamin D?

Authors’ response: Vitamin D deficiency is commonly observed in patients with obesity and diabetes, who carry an excess risk for poor clinical outcomes upon SARS-CoV2 infection. The antiviral effects of vitamin D have been attributed both to a direct interference with viral replication and to immunomodulatory and anti-inflammatory effects. The role of vitamin D in preventing COVID-19 infection and progression is currently not clear. A negative correlation has been reported between vitamin D levels and the prevalence of COVID-19 in European countries. However, the existing evidence on the association between vitamin D status and COVID-19 severity/mortality remains insufficient. We provide all these new data on the role of vitamin D for COVID-19 on page 23, lines 654-672, as suggested by the reviewer.  

  1. There is discussion on a number of drugs and COVID - what is the situation with COVID and ibuprofen?

Authors’ response: Current epidemiological evidence is not strong enough to support a causal link between ibuprofen use and poor outcomes in COVID-19. We discuss this controversy in a whole new subsection of section 5.2.3 entitled “Non-steroidal anti-inflammatory drugs (Ibuprofen)” on page 22, lines 611-623 of the revised manuscript. 

  1. There is no consistency with blood glucose levels - mg/dl need to be provided with mmol/L all through the text.

Authors’ response: We provide blood glucose levels in mmol/l consistently throughout the revised manuscript, as suggested by the reviewer through the whole manuscript. We also report HbA1c values as both percentages (as Diabetes Control and Complications Trial units) and as mmols/mol values (as International Federation of Clinical Chemistry units).

Thank you very much for your time

Sincerely,

Professor N. Tentolouris, MD

On behalf of the co-authors

Reviewer 2 Report

The current review represents a well written and comprehensive manuscritpt-review about COVID-19 and Diabetes Mellitus. I have however some comments that I think are important to be addressed:

  • Although this manuscript is a review article , Methods section is missing and need to be added. That will provide some information of what was the focused of the review article, how the authors collected the articles and until when they collected articles.
  • Authors make some strong statements, which maybe true, however they are not supported with the current evidence/data. For example: We are not sure that DM is an independent factor. DM patients have indeed a higher risk for COVID19 however is it truly independent? As frequently DM patients have comorbidities secondary to their DM complications (CAD,CKD etc) ?

Consider also to add another recent published study Shi Q, Zhang X, Jiang F, et al. Clinical characteristics and risk factors for mortality of COVID-19 patients with diabetes in Wuhan, China: a two-center, retrospective study. Diabetes Care 2020;43:1382–1392

  • Another strong statement that needs to be revised is whether improving glycemic control can lead to better outcomes in patients with COVID 19 and DM. In the hospital data are limited. This recently published study supports the statements of the authors Celestino Sardu, Nunzia D’Onofrio, Maria Luisa Balestrieri, Michelangela Barbieri, Maria Rosaria Rizzo, Vincenzo Messina, Paolo Maggi, Nicola Coppola, Giuseppe Paolisso, Raffaele Marfella Diabetes Care 2020 May; 

Previous studies in non COVID 19 patients showed the opposite : It lead to increased mortality (NICE sugar).  So I think overall we do not have the evidence yet to make such strong recommendations about tight glycemic control

  • CGM is a novel way to measure glucose values. In the hospital it is still considered investigational, or at least it has not been recommended to replace the POC and become a standard of care technology. Due to the pandemic situation, CGM devices have been used in order to help physicians/nurses. Using CGM devices the “traditional” way carries still a risk of transmission as there is no decreased direct contact. Remote monitoring with CGM devices has been described a couple of years ago using DEXCOM devices and also with Medtronic systems recently.  This is how the CGM systems can be used in a better way to reduce PPE exposure and risk of COVId19 transmission 
  • SGLT2, TZDs: For both these medications we do not have any evidence of safety ir efficacy to use them in the hospital setting . Using SGLT2 in someone with a potential severe infection should not be appropriate. These medications reduce intravascular volume and therefore should be avoided in any infectious disease. And as )more importantly) there are no evidence to use them in the hospital (in general) I would avoid to  use them 

Author Response

Authors’ Reply to Reviewer #2 Comments

We thank the reviewer for the critical input and the helpful suggestions, which helped us revise and improve the quality of our manuscript. We considered all comments, performed all necessary revisions and we provide below a detailed point-by-point response to every single comment, hoping to have adequately addressed all concerns. All changes are denoted with red colour in the revised version of our manuscript.

  1. Although this manuscript is a review article, Methods section is missing and needs to be added. That will provide some information of what was the focus of the review article, how the authors collected the articles and until when they collected articles.

Authors’ response: In the revised version of our manuscript, we have added a new section after Introduction entitled “Literature search strategy and selection criteria” (page 4, section 2, lines 109-119), as suggested by the reviewer. In this section, we briefly describe the methodology of literature search and data collection and the main focus of our review article.

  1. Authors make some strong statements, which may be true, however they are not supported with the current evidence/data. For example: We are not sure that DM is an independent factor. DM patients have indeed a higher risk for COVID19 however is it truly independent? As frequently DM patients have comorbidities secondary to their DM complications (CAD, CKD etc)?

Authors’ response: We absolutely agree with the reviewer that it is difficult to disentangle whether the association of DM with COVID-19 is truly independent or rather mediated by coexisting DM-related comorbidities such as cardiovascular disease, hypertension, obesity or chronic kidney disease. Issues of heterogeneity, poor reporting and the lack of high-quality systematic reviews make it difficult to definitively conclude whether DM represents indeed an independent risk factor for adverse COVID-19 clinical outcomes. To better emphasize this unresolved issue, we refer to this uncertainty in section 6 (Knowledge gaps and questions to be answered, page 24, second bullet, lines 692-693), and we rephrased our strong statements regarding independent association into more moderate ones throughout the manuscript (i.e. page 10, line 291-292). 

  1. Consider also to add another recent published study Shi Q, Zhang X, Jiang F, et al. Clinical characteristics and risk factors for mortality of COVID-19 patients with diabetes in Wuhan, China: a two-center, retrospective study. Diabetes Care 2020;43:1382–1392.

Authors’ response: In the new reference cited by the reviewer, DM was not independently associated with in-hospital mortality in patients with COVID-19, and furthermore, advanced age and comorbid hypertension were identified as independent predictors of mortality in patients with DM. In response to the reviewer’s comment, we present these new data on page 6, section 3.2 (Clinical course of SARS-CoV2 infection in patients with DM), lines 184-186 (independent predictors of mortality in patients with DM), and page 6, lines 191-192 (lack of independent association of DM with mortality in COVID-19 patients).

  1. Another strong statement that needs to be revised is whether improving glycemic control can lead to better outcomes in patients with COVID 19 and DM. In the hospital data are limited. This recently published study supports the statements of the authors Celestino Sardu, Nunzia D’Onofrio, Maria Luisa Balestrieri, Michelangela Barbieri, Maria Rosaria Rizzo, Vincenzo Messina, Paolo Maggi, Nicola Coppola, Giuseppe Paolisso, Raffaele Marfella Diabetes Care 2020 May; Previous studies in non COVID 19 patients showed the opposite: It lead to increased mortality (NICE sugar). So I think overall we do not have the evidence yet to make such strong recommendations about tight glycemic control.

Authors’ response: The reviewer is correct, although there are convincing data to suggest that improving glycemic control can lead to better outcomes during infections in patients with diabetes, the information with COVID-19 is still limited. As the reviewer points out, the NICE SUGAR trial gave the final verdict that blood glucose levels during hospitalization in these subjects should be kept between 4.4-10 mmol/l in order to avoid hypoglycaemia, which would increase the risk of death; this is pointed out in section 3.4, page 12, lines 365-372). The recent study by Sardu et al. in Diabetes Care is indeed important and supports the earlier suggestions with regards to glycemic control; this study has been included and discussed in our paper (section 3.4, page 13, lines 400-415). In addition, according to the reviewer’s suggestion, we revised our strong statement about the importance of glycemic control as follows: (a) we replaced “is” by “may be” and “can” by “could be” in several parts of this section (page 11, line 319; page 13, lines 391, 392, 397), (b)  we added a sentence to indicate that data with SARS-CoV2 are still limited (page 13, lines 498-399), (c)  We added a brief conclusion that glycemic control may be protective during infection by COVID-19, as long as hypoglycemia is avoided (page 13, lines 411-415).

  1. CGM is a novel way to measure glucose values. In the hospital it is still considered investigational, or at least it has not been recommended to replace the POC and become a standard of care technology. Due to the pandemic situation, CGM devices have been used in order to help physicians/nurses. Using CGM devices the “traditional” way carries still a risk of transmission as there is no decreased direct contact. Remote monitoring with CGM devices has been described a couple of years ago using DEXCOM devices and also with Medtronic systems recently. This is how the CGM systems can be used in a better way to reduce PPE exposure and risk of COVID19 transmission.

Authors’ response: We reproduce all this interesting information regarding CGM provided by the reviewer on pages 15-16, subsection 5.2.2 (Glucose monitoring), lines 480-485 of the revised manuscript.

  1. SGLT2, TZDs: For both these medications we do not have any evidence of safety or efficacy to use them in the hospital setting. Using SGLT2 in someone with a potential severe infection should not be appropriate. These medications reduce intravascular volume and therefore should be avoided in any infectious disease. And as (more importantly) there is no evidence to use them in the hospital (in general), I would avoid using them.

Authors’ response: We absolutely endorse the reviewer’s opinion that both medications (SGLT2 inhibitors and TZDs) should be avoided in severely infected patients in the hospital setting due to the concerns of intravascular volume depletion and fluid retention, respectively. The reviewer is correct that the use of these agents in hospitalized patients with infection is not appropriate. In the revised manuscript, we emphasize these safety concerns using the clear-cut statement “There is no evidence of safety or efficacy for use in the hospital setting” suggested by the reviewer at the beginning of both sections (page 19, line 525 for SGLT2 inhibitors and page 20, line 576 for TZDs). Furthermore, for SGLT2 inhibitors, we added the strong statement “Their use in patients with severe infection is not appropriate” (page 19, lines 527-528) to avoid any confusion. In Table 1, it is explicitly stated that both medications should be discontinued in severe hospitalized cases due to important safety concerns (page 18, Table 1).

Thank you very much for your time

Sincerely,

Professor N. Tentolouris, MD

On behalf of the co-authors